# TRAP1 Is Expressed in Human Retinal Pigment Epithelial Cells and Is Required to Maintain their Energetic Status

**DOI:** 10.3390/antiox12020381

**Published:** 2023-02-04

**Authors:** Inês Ramos Rego, Daniela Silvério, Maria Isabel Eufrásio, Sandra Sofia Pinhanços, Bruna Lopes da Costa, José Teixeira, Hugo Fernandes, Yang Kong, Yao Li, Stephen H. Tsang, Paulo J. Oliveira, Rosa Fernandes, Peter M. J. Quinn, Paulo Fernando Santos, António Francisco Ambrósio, Celso Henrique Alves

**Affiliations:** 1Coimbra Institute for Clinical and Biomedical Research (iCBR), Faculty of Medicine, University Coimbra, 3000-548 Coimbra, Portugal; 2Center for Innovative Biomedicine and Biotechnology (CIBB), University Coimbra, 3004-504 Coimbra, Portugal; 3Clinical Academic Center of Coimbra (CACC), 3000-548 Coimbra, Portugal; 4Faculty of Sciences and Technology, University Coimbra, 3030-790 Coimbra, Portugal; 5CNC-Center for Neuroscience and Cell Biology, University of Coimbra, UC Biotech, Parque Tecnológico de Cantanhede, 3060-197 Coimbra, Portugal; 6Department of Biomedical Engineering, The Fu Foundation School of Engineering and Applied Science, Columbia University, New York, NY 10027, USA; 7Department of Ophthalmology, Vagelos College of Physicians & Surgeons, Columbia University, New York, NY 10032, USA; 8Faculty of Medicine, University Coimbra, 3000-370 Coimbra, Portugal; 9Jonas Children‘s Vision Care, and Bernard and Shirlee Brown Glaucoma Laboratory, Columbia Stem Cell Initiative, Pathology and Cell Biology, Institute of Human Nutrition, Vagelos College of Physicians and Surgeons, Columbia University, New York, NY 10032, USA; 10Association for Innovation and Biomedical Research on Light and Image (AIBILI), 3000-548 Coimbra, Portugal; 11Institute of Pharmacology and Experimental Therapeutics, Faculty of Medicine, University Coimbra, 3000-548 Coimbra, Portugal; 12Department of Life Sciences, University Coimbra, 3000-456 Coimbra, Portugal

**Keywords:** age-related macular degeneration (AMD), retinal pigment epithelium (RPE), mitochondria, oxidative stress, tumor necrosis factor receptor-associated protein 1 (TRAP1)

## Abstract

Age-related macular degeneration (AMD) is the leading cause of severe vision loss and blindness in elderly people worldwide. The damage to the retinal pigment epithelium (RPE) triggered by oxidative stress plays a central role in the onset and progression of AMD and results from the excessive accumulation of reactive oxygen species (ROS) produced mainly by mitochondria. Tumor necrosis factor receptor-associated protein 1 (TRAP1) is a mitochondrial molecular chaperone that contributes to the maintenance of mitochondrial integrity by decreasing the production and accumulation of ROS. The present study aimed to evaluate the presence and the role of TRAP1 in the RPE. Here, we report that TRAP1 is expressed in human adult retinal pigment epithelial cells and is located mainly in the mitochondria. Exposure of RPE cells to hydrogen peroxide decreases the levels of TRAP1. Furthermore, TRAP1 silencing increases intracellular ROS production and decreases mitochondrial respiratory capacity without affecting cell proliferation. Together, these findings offer novel insights into TRAP1 functions in RPE cells, opening possibilities to develop new treatment options for AMD.

## 1. Introduction

Age-related macular degeneration (AMD) is the major cause of serious vision loss and blindness in seniors worldwide [1]. Accumulation of drusen in the macula and pigmentary abnormalities are hallmarks of the early stages of AMD [2]. Progression to late AMD is characterized by the loss of both retinal pigment epithelium (RPE) and photoreceptor cells (dry AMD) or abnormal proliferation of choroidal capillaries (wet AMD) [2,3]. The RPE is a monolayer of polarized and pigmented cells located between the photoreceptor cells and the choroid, and it is an essential component of the outer blood–retinal barrier [4,5]. The RPE plays an important role in the maintenance of overlying photoreceptor cells by performing crucial functions such as transepithelial transport, phagocytosis of photoreceptor outer segments, and protection against oxidative stress [4]. Degeneration and impaired clearance mechanisms of RPE contribute to increased accumulation of lipofuscin, which is a yellowish aggregate of oxidized proteins and lipids, the main constituent of drusen, and one of the hallmarks of AMD. Therefore, RPE has been suggested as a critical site of pathology [6]. Retinal pigment epithelial cells are particularly susceptible to oxidative stress due to exposure to intense focal light, high metabolic activity, unique phagocytic function, large oxygen gradient from the choroid, and accumulation of oxidized lipoproteins with ageing [6]. In the context of AMD, oxidative-stress-induced RPE damage results from the excessive accumulation of reactive oxygen species (ROS), which is produced mainly in the mitochondria under pathological conditions. Mitochondrial function homeostasis is critical for maintaining ROS at physiological levels, avoiding the metabolic dysfunction observed in AMD pathology [7]. Therefore, it is important to study novel proteins that might be involved in oxidative stress mechanisms in the RPE in the context of AMD. Tumor necrosis factor receptor-associated protein 1 (TRAP1), also known as heat shock protein 75 (HSP75), is a mitochondrial molecular chaperone that supports protein folding and contributes to the maintenance of mitochondrial integrity, even under cellular stress [8,9,10]. TRAP1 has attracted increasing interest for its homology to heat shock protein 90 (HSP90), and it has long been pursued as a potential therapeutic target for designing novel anticancer agents [9,11,12,13]. Previous studies in several cell lines demonstrated that TRAP1 protects against mitophagy, mitochondrial apoptosis, and dysfunction by decreasing the production and accumulation of ROS [8,14,15,16,17]. It has also been suggested that downregulation of TRAP1 expression impacts cellular function, resulting in lower mitochondrial membrane potential, increased intracellular ROS production and increased cell death [18,19,20]. TRAP1 inhibits mitochondrial oxidative phosphorylation (OXPHOS) via downregulation of cytochrome oxidase in the respiratory chain [9]. Considering the above-mentioned TRAP1 functions described in other cell types [8,9,10,11,12,13,14,15,16,17,18,19,20], we hypothesize that TRAP1 modulates mitochondrial metabolism in the RPE, and that it has a role in avoiding AMD onset and progression. The present study aimed to evaluate the presence and function of TRAP1 in the RPE and whether TRAP1 mediates protection against oxidative stress in RPE cells. Here, we report that TRAP1 is expressed in human adult RPE cells and located mainly in the mitochondria. Furthermore, TRAP1 silencing increased ROS production by human RPE cells, ultimately driving cells into a quiescent metabolic state. Taken together, these results provide novel insights into TRAP1 antioxidant functions in RPE cells and the susceptibility of RPE to oxidative stress.

## 2. Materials and Methods

### 2.1. Cell Culture of Human RPE Cells

The human retinal pigment epithelial cell line ARPE-19 was obtained from the American Type Culture Collection (ATCC^®^ CRL-2302, Manassas, VA, USA). The cells (passages 6 to 35) were maintained in Dulbecco’s Modified Eagle Medium/Nutrient Mixture F-12 (DMEM/F-12) with GlutaMAX™ (Gibco, Waltham, MA, USA) supplemented with 10% (*v*/*v*) Fetal Bovine Serum (FBS) (Gibco, Waltham, MA, USA) and antibiotics (100 U/mL penicillin, 100 μg/mL streptomycin (Gibco, Waltham, MA, USA) under a humidified atmosphere of 5% CO_2_ at 37 °C. 

### 2.2. Cell Culture of Human Adult Donor RPE, iPSC-Derived RPE (iRPE) Cells and iPSC-Derived RPE Spheroids

Donor native RPE was isolated from a human autopsy eye shell purchased from the Eye-Bank for Sight Restoration (New York, NY, USA). Previously validated and published wild-type hiPSC lines [21,22] were differentiated at passages 3 to 6 according to a previous publication [23]. In brief, iPSCs were cultured to confluence in 6-well culture dishes pre-treated with 1:50 diluted Matrigel (Corning, New York, NY, USA) in Knock-Out (KO) DMEM-based differentiation medium with 10 mmol/L nicotinamide (Sigma-Aldrich, St. Louis, MO, USA) for the first 14 days. During the 3rd to 4th weeks of differentiation, the differentiation medium was supplemented with 100 ng/mL human Activin-A (Peprotech, Cranbury, NJ, USA). From day 29, Activin-A was removed until differentiation was completed. After 8–10 weeks, pigmented clusters formed and were manually picked and plated on Matrigel-coated dishes. Those cells were maintained in RPE medium as previously described [24]. The iRPE cells were cultured for another 6–8 weeks to allow them to form a functional monolayer for functional assays. Spheroids of hiPSC-derived RPE were obtained from the differentiation of retinal organoids as previously reported [24,25,26]. Previously validated and published wild-type hiPSC lines [21,22] and a commercially obtained Tom20-mEGFP hiPSC line (AICS-0011 cl.27) were maintained on Matrigel (BD)-coated plates in mTeSR plus medium (STEMCELL Technologies, Vancouver, Canada) and passaged with ReleSR (STEMCELL Technologies, Vancouver, Canada). Retinal organoid differentiation was carried out using the agarose microwell array seeding and scraping method with minor modifications [27,28]. In brief, 90% confluent iPSCs were detached with ReleSR (STEMCELL Technologies, Vancouver, Canada). Cells were counted and seeded at 2000 cells per microwell and incubated with (±) blebbistatin in mTeSR plus medium overnight and transitioned from mTeSR plus to Neural Induction Medium 1 (NIM)-1 over the next 3 differentiation days (DD), forming embryoid bodies (EBs). On DD7, EBs were moved to Matrigel-coated wells until DD28. On DD16, the medium was transitioned from NIM-1 to NIM-2. Neuroepithelia were lifted using the checkerboard-scrapping method. The lifted retinal organoids were maintained until DD41 in NIM-2 on poly-HEMA (Sigma-Aldrich, St. Louis, USA)-coated wells. Retinal lamination medium 1 (RLM-1) was used from DD42 to DD69. RLM-2 was used from DD70 to DD97. RLM-3 was used from DD98 for long-term culture. NIM1 (50 mL): 48.95 mL DMEM/F12, 10 μL 10 mg/mL heparin (final concentration, 2 μg/mL), 0.5 mL Media-Non Essential Amino Acids (100×, MEM NEAA), 0.5 mL N2 supplement (100×). NIM2 (50 mL): 48 mL DMEM/F12 (3:1), 0.5 mL MEM NEAA, 1 mL B27 Supplement (50×, minus vitamin A), 0.5 mL penicillin-streptomycin (P/S, 10,000 U/mL). RLM1 (50 mL): 42.9 mL DMEM/F12 (3:1), 0.1 mL taurine (100 μM final concentration), 5 mL FBS, 1 mL B27, 0.5 mL MEM NEAA, 0.5 mL P/S. 15. RLM2: RLM1 supplemented with 0.1 μL per mL of 10 mM retinoic acid. RLM3: RLM1 without B27, replaced with N2 supplement and retinoic acid reduced to 0.05 μL per mL.

### 2.3. Immunofluorescence

ARPE-19 cells were seeded in DMEM/F-12 with GlutaMAX™ (Gibco, Waltham, MA, USA) supplemented with 1% FBS (Gibco, Waltham, MA, USA) and antibiotics (100 U/mL penicillin, 100 μg/mL streptomycin) (Gibco, Waltham, MA, USA) at a density of 60,000 cells/well on 12 mm glass coverslips in 24 well-plates for 24 h before treatment. After the respective treatment, cells were fixed with 4% Paraformaldehyde (PFA) with 4% sucrose for 10 min, washed in PBS, and blocked for 1 h with a solution containing 10% normal goat serum, 3% bovine serum albumin, and 0.3% Triton X-100 in PBS. Afterwards, cells were incubated with the primary antibodies for 1 h at room temperature. The primary antibodies used were TRAP1 (1:200; BD Bioscience, Franklin Lakes, NJ, USA), SAM50 (1:200; Sigma-Aldrich, St. Louis, USA) and Ki67 (1:100, Agilent, Santa Clara, CA, USA). After washing with PBS, cells were incubated for 1 h at room temperature with the fluorescent-labelled secondary antibodies: goat anti-mouse/rabbit Alexa Fluor 488 (1:200; Invitrogen, Waltham, MA, USA) and goat anti-mouse/rabbit Alexa Fluor 568 (1:200; Invitrogen, Waltham, MA, USA) and counterstained with DAPI (1:5000; Invitrogen, Waltham, MA, USA). Samples were mounted with Glycergel mounting medium (Dako Cytomation, Glostrup, Denmark) and imaged using a confocal microscope (Zeiss LSM510, Carl Zeiss, Oberkochen, Germany). DD100 iPSC-derived RPE spheroids (*n* = 5) were washed twice in PBS before being fixed with 4% PFA for 30 min, and then cryoprotected with 15% followed by 30% sucrose. Subsequently, iPSC-derived RPE were embedded in Tissue-Tek O.C.T. Compound (Sakura Finetek, Torrance, Canada) and cryosectioned. Cryosections were rehydrated in PBS and blocked for 1 h with a solution containing 10% normal goat serum, 1% bovine serum albumin (BSA), and 0.4% Triton X-100 in PBS. The TRAP1 (1:200; BD Bioscience, Franklin Lakes, USA) antibody was diluted in 0.3% normal goat serum, 0.4% Triton X-100 and 1% BSA in PBS and incubated overnight at 4 °C. Sections were washed with PBS and then incubated with a fluorescent goat anti-mouse Alexa Fluor 555 (1:1000) secondary antibody diluted in 1% BSA in PBS for 1 h at room temperature. Samples were counterstained and mounted using VECTASHIELD Vibrance Antifade Mounting Medium with DAPI (H-1800, Vector Laboratories, Burlingame, CA, USA) and imaged using a confocal microscope (Nikon Ti Eclipse inverted microscope, Nikon, Tokyo, Japan).

### 2.4. Polymerase Chain Reaction

ARPE-19 cells were washed twice with ice-cold sterile PBS and total RNA was extracted using the Trizol^®^ reagent (Life Technologies, Carlsband, CA, USA) according to the manufacturer’s protocol. RNA concentration and purity were determined using NanoDrop^®^ (Thermo Fisher Scientific, Waltham, MA, USA). One μg of total RNA was reverse-transcribed into cDNA using the NZY First-Strand cDNA Synthesis Kit (NZYTech, Lisbon, Portugal) according to the manufacturer’s instructions. The cDNA was then treated with RNase-H for 20 min at 37 °C and stored at −20 °C until further analysis. Amplification of the cDNA was performed using a BioRad T100 Thermal Cycler (Bio-Rad, Hercules, CA, USA) in a total volume of 40 μL containing MYtaq Red mix (NZYTech, Lisbon, Portugal) and 0.25 mM of specific primers for TRAP1: HA29_TRAP1_FW 5′-GCTCTGGGAGTACGACATGG-3′ and HA30_TRAP1_RV 5′-CTCCGAGGACACAGAATTGGT-3′ (129 bp amplicon). Cycling conditions were as follows: melting step at 95 °C for 1 min, annealing at 62 °C for 30 s and elongation at 72 °C for 30 s, for 35 cycles. Total RNA from iRPE, donor RPE and RPE spheroids (DD114) was harvested using the RNeasy kit (Qiagen, Hilden, Germany) according to the manufacturer’s instructions. DNase I (Invitrogen, Waltham, USA) treatment was conducted to avoid genomic DNA contamination. The reverse transcription reaction was performed using the SuperScript™ III First-Strand Synthesis SuperMix (Thermo Fisher Scientific, Waltham, MA, USA) to generate cDNA. PCR was performed using Phire Hot Start II DNA Polymerase (Thermo Fisher Scientific, Waltham, USA) to determine gene expression. The PCR program was 25 °C for 10 min, 50 °C for 30 min, 85 °C for 5 min and 4 °C for 5 min. The primers used for PCR were HA29_TRAP1_FW 5′-GCTCTGGGAGTACGACATGG-3′ and HA30_TRAP1_RV 5′-CTCCGAGGACACAGAATTGGT-3′. PCR product was separated using a 2% (*w*/*v*) agarose gel. 

### 2.5. Western Blotting

ARPE-19 protein extracts were prepared in ice-cold radioimmunoprecipitation assay (RIPA) buffer with 1 mM of dithiothreitol (Sigma-Aldrich, St. Louis, MO, USA) and a protease inhibitor cocktail (Sigma-Aldrich, St. Louis, MO, USA). The protein concentration was measured by Bradford’s assay (Invitrogen, Waltham, MA, USA). Forty μg of total protein lysates were resolved on a 12% SDS-PAGE gel and transferred onto a PVDF membrane. The membranes were then blocked in 5% skim milk for 1 h and incubated for 16 h at 4 °C with primary anti-TRAP1 (1:1000; BD Bioscience, Franklin Lakes, NJ, USA) and anti-β-actin (1:5000; Sigma-Aldrich, St. Louis, MO, USA). After washing 3 times with Tris Buffered Saline with 0.1% Tween 20 (TBST) for 10 min, the membranes were incubated for 1 h at room temperature with the respective secondary antibody (anti-mouse, 1:10,000; Invitrogen, Waltham, MA, USA) and washed 3 times with TBST for 10 min each. After that, membranes were imaged in ImageQuant LAS 500 (GE Healthcare Bio-Sciences, Chicago, IL, USA) using WesternBright Sirius™ (Advansta, San Jose, CA, USA). The protein band densitometry was quantified using the software Quantity One (Bio-Rad, Hercules, CA, USA), using β-actin as a loading control. 

Western blot analysis of RPE organoids (DD408) was performed based on a previously published method with minor modifications [29]. In brief, RPE organoids were directly lysed in 100 μL M-PER Mammalian Protein Extraction Reagent (Thermo Fisher Scientific, Waltham, MA, USA) with protease inhibitors (Sigma-Aldrich, St. Louis, MO, USA). After sonication, the lysates were centrifuged for 10 min at 20,000× *g* and 4 °C. Then, 75 μL lysates were mixed with 25 μL of 4× Laemmli sample buffer (Bio-Rad, Hercules, USA) containing 10% (*v*/*v*) of 2-mercaptoethanol (Sigma-Aldrich, St. Louis, USA) and incubated at 95 °C for 10 min. Samples were separated using a 4–15% Bis-Tris gel (Bio-Rad, Hercules, CA, USA), and transferred onto nitrocellulose membranes (Bio-Rad, Hercules, CA, USA), followed by 2 h of blocking in 5% (*w*/*v*) non-fat milk in PBS containing 0.1% (*v*/*v*) Tween 20 (PBST). The membranes were incubated with mouse TRAP1 primary antibody (1:1000; BD Bioscience, Franklin Lakes, NJ, USA) or mouse β-Actin primary antibody (1:2000; Abcam, Waltham, MA, USA) diluted in 5% (*w*/*v*) non-fat milk in PBST overnight at 4 °C. After overnight incubation, the membranes were washed three times with PBST for 10 min each and then incubated with secondary antibody (rabbit anti-mouse IgG-HRP antibody—1:5000; Abcam, Waltham, MA, USA) for 1 h at room temperature. After washing the membrane three times with PBST for 10 min each, protein bands were visualized manually using the Med-Dent ready-to-use developer/fixer combo (Z&Z Medical, Cedar Falls, IA, USA) after exposure to Immobilon western chemiluminescent substrate (Millipore, Burlington, VT, USA).

### 2.6. TRAP1 Silencing in ARPE-19 Cells 

To silence TRAP1 expression, we used a siRNA anti-Human TRAP1 (siTRAP1, NM_016292, SI00115150 A60-SIRNA, Qiagen, Hilden, Germany) [30]. As a negative control, a scramble siRNA (siCTL, Unspecific_AllStars_1 A60-SIRNA SI03650318; Qiagen, Hilden, Germany) was used. On day 0, ARPE-19 cells were seeded in DMEM/F-12 with GlutaMAX™ supplemented with 1% FBS and antibiotics (100 U/mL penicillin, 100 µg/mL streptomycin) at a density of 10,000, 60,000 or 115,000 cells/well in a 96-, 24- or 6-well-plate, respectively, to be approximately 80% confluent at the day of transfection. On day 1, 4 h before the transfection, the medium was replaced by Opti-MEM medium (Gibco, Waltham, MA, USA). Afterwards, lipofectamine RNAiMAX transfection reagent (Invitrogen, Waltham, MA, USA) was diluted in Opti-MEM medium. The master mix containing 0.1 µM of siCTL or siTRAP1 diluted in Opti-MEM medium was added to the diluted RNAiMAX in a 1:1 ratio and incubated for 5 min at room temperature. Thereafter, 10, 50 or 250 μL of the previous solution, depending on the well size, was added dropwise to the cells. Cells were analyzed at 24 or 72 h after transfection. In the cells cultured for 72 h, the culture medium was replaced by DMEM/F-12 with GlutaMAX™ supplemented with 1% FBS and antibiotics 48 h after transfection.

### 2.7. Resazurin Assay

ARPE-19 cells were seeded (10,000 cells/well) in flat-bottom 96-well microplates in DMEM/F-12 with GlutaMAX™ supplemented with 1% FBS and antibiotics (100 U/mL penicillin, 100 g/mL streptomycin), at a density of 10,000 cells/well in flat-bottom 96-well microplates. ARPE-19 cells were then incubated with 31.25, 62.5, 125, 500, 1000 or 2000 µM of H_2_O_2_ (PanReac AppliChem, Ottoweg, Germany) for 24 h. In the experiments to assess the impact of TRAP1 loss in ARPE-19, the cells were transfected twenty-four hours after seeding, as previously described. Cell viability/metabolism was assessed using Resazurin (Invitrogen, Waltham, MA, USA) according to the manufacturer’s instructions. Briefly, the medium was replaced with DMEM/F-12 with GlutaMAX™ containing antibiotics and resazurin (1 mg/mL) and incubated at 37 °C for 2 h. The fluorescence was measured at Ex/Em = 550/590 nm using Synergy Multi-Mode Reader (BioTek, Winooski, VT, USA). Cells were then washed twice with PBS and fixed with 4% PFA in PBS for 10 min. Thereafter, nuclei were counterstained with DAPI (1/5000; Invitrogen, Waltham, MA, USA) and counted using an automated high-content fluorescence microscope (INCell Analyzer 2000, GE Healthcare, Chicago, IL, USA). The values are presented as a percentage of the control (non-treated cells).

### 2.8. Sulforhodamine B Assay

Seventy-two hours after transfection, the culture medium was removed, and cells were washed with 1% phosphate-buffered saline (PBS). The cells were then fixed with ice-cold 1% acetic acid in 100% methanol for 16 h at −20 °C. The microplates containing fixed cells were dried at 37 °C. Thereafter, cells were incubated with 0.5% Sulforhodamine B (SRB) (Sigma-Aldrich, St. Louis, MO, USA) in 1% acetic acid for 30 min at 37 °C and subsequently washed three times with 1% acetic acid. Plates were then allowed to dry before adding 10 mM Tris (pH 10), followed by incubation at room temperature for 15 min with agitation. Lastly, 200 μL of the solubilized solution was added to each well and optical density was measured at 540 nm.

### 2.9. Dihydroethidium Probe

Superoxide anion was detected using the Dihydroethidium (DHE) probe. ARPE-19 cells were seeded in DMEM/F-12 with GlutaMAX™ supplemented with 1% FBS and antibiotics (100 U/mL penicillin, 100 µg/mL streptomycin) at a density of 60,000 cells/well in 12 mm glass coverslips placed on a 24-well plate. TRAP1 silencing was performed as previously described. Twenty-four or seventy-two hours after transfection, cells were incubated with 10 μM of Dihydroethidium (Invitrogen, Waltham, USA) diluted in PBS with 5% FBS for 30 minutes at 37 °C in a light-protected environment. Thereafter, cells were washed twice with PBS, fixed with 4% PFA with 4% sucrose for 10 min and permeabilized with PBS containing 1% Triton X-100 for 5 min. Cells were then counterstained with DAPI (1:5000; Invitrogen, Waltham, USA) diluted in PBS containing 10% normal goat serum, 3% bovine serum albumin and 0.3% Triton X-100. Samples were mounted with Glycergel mounting medium (Dako Cytomation, Glostrup, Denmark) and imaged using a confocal microscope (Zeiss LSM510, Carl Zeiss, Oberkochen, Germany). Mean fluorescence intensity (MFI) and the number of DAPI-positive nuclei were automatically calculated/counted using ImageJ v1.53C, using at least 4 images for each condition/experiment. The values are presented as MFI per cell. 

### 2.10. Transmission Electron Microscopy

Seventy-two hours after transfection with siCTL or siTRAP1, ARPE-19 cells were fixed with 2.5% glutaraldehyde in 0.1 M sodium cacodylate buffer (pH 7.2) supplemented with 1 mM calcium chloride. Sequential post-fixation was performed using 1% osmium tetroxide (Sigma-Aldrich, St. Louis, MO, USA) for 1 h and contrast-enhanced with 1% aqueous uranyl acetate for 90 min. After rinsing in distilled water, samples were dehydrated in a graded ethanol series (50–100%), impregnated, and embedded using an Epoxy embedding kit (Fluka Analytical, Waltham, MA, USA). Ultrathin sections (70 nm) were mounted on copper grids (300 mesh) and stained with 0.2% lead citrate for 7 min. Observations were carried out using an FEI-Tecnai G2 Spirit Bio Twin at 100 kV. 

### 2.11. Mitochondrial Network

Mitochondria were labelled with Image-iT™ TMRM Reagent (Invitrogen, Waltham, MA, USA). Briefly, 72 h after transfection, ARPE-19 cells were incubated with 75 nM TMRM Reagent diluted in Hank’s Balanced Salt Solution (HBSS) containing Ca^2+^ and Mg^2+^ for 30 min at 37 °C. Thereafter, the TMRM Reagent solution was removed and replaced by Opti-MEM medium. Cell images were acquired using a Zeiss LSM710 confocal microscope (Carl Zeiss, Oberkochen, Germany) equipped with an environmental chamber. Analysis of the images was performed using ImageJ v1.53K (National Institute of Health, Bethesda, MD, USA) software. Parameters were calculated using the Mito-Morphology ImageJ macro [31]. The average area/perimeter ratio was employed as an index of mitochondrial interconnectivity and inverse circularity used to assess mitochondrial elongation [31].

### 2.12. Oxygen Consumption and Extracellular Acidification Rate

The oxygen consumption rate (OCR) and extracellular acidification rate (ECAR) of ARPE-19 cells were determined using Seahorse XFe96 Extracellular Flux Analyzer Technology (Agilent, Santa Clara, CA, USA). Cells (10,000 per well) were seeded in XF96–well plates (Agilent, Santa Clara, CA, USA). Cells were transfected 24 h later with siCTL or siTRAP1 and cultured for another 48 h. The cell culture medium was then replaced by 175 µL/well of pre-warmed low-buffered serum-free minimal DMEM medium (D5030; Sigma-Aldrich, St. Louis, USA) supplemented with glucose (17.5 mM), L-glutamine (2 mM) and sodium pyruvate (0.5 mM). The pH was adjusted to 7.4 and the cells incubated at 37 °C in a non-CO_2_ incubator for 1 h to allow the temperature and pH of the medium to reach equilibrium before the first-rate measurement. Oligomycin, FCCP, rotenone, and antimycin A solutions were prepared in DMSO, diluted in low-buffered serum-free DMEM medium, and then pre-loaded (25 µL) into the ports of each well in the XFe96 sensor. The sensor cartridge and the calibration plate were loaded into the XFe96 Extracellular Flux Analyzer for calibration. Then, 2 µM oligomycin was injected into reagent delivery port A, 1 µM FCPP was injected into port B, and 1 µM rotenone plus 1µM antimycin A were injected into reagent delivery port C and mixed. For oxygen consumption rate (OCR) measurements, three baseline rate measurements of OCR and extracellular acidification rate (ECAR) of the cells were made using a 3 min mix and a 5 min measure cycle. The total amount of protein was determined using the Sulforhodamine B (SRB) assay, and each OCR’s raw data was normalized to the total protein content in each well. Results were analyzed by using the Software Version Wave Desktop 2.6 (Agilent, Santa Clara, CA, USA).

### 2.13. Statistical Analysis

The normality of the distribution was tested by the Kolmogorov–Smirnov test. Statistical significance was calculated by using an unpaired t-test or by using the Mann–Whitney U test if the data did not follow a normal distribution. All statistical analyses were performed using GraphPad Prism, version 9.0 (GraphPad Software, Boston, MA, USA). All values are expressed as mean ± standard error of the mean (SEM). For all tests, *p* values < 0.05 were considered statistically significant. At least three independent experiments were used for analysis (*n* = 3), unless otherwise stated.

## 3. Results

### 3.1. TRAP1 Localizes in the Mitochondria of Human RPE Cells

The expression of TRAP1 mRNA in ARPE-19 cells was confirmed by PCR analysis using specific primers (Figure 1A), while TRAP1 protein was also detected in these cells (Figure 1B–D). Furthermore, we showed that TRAP1 protein is located in the mitochondrial compartment since its expression co-localized with MitoTracker (Figure 1C), a dye that accumulates specifically in mitochondria [32], and with SAM50 (Figure 1D), a mitochondrial protein. We further confirmed the presence of TRAP1 mRNA in the human RPE cells using samples from human induced pluripotent stem cells (hiPSCs)-derived RPE (iRPE), RPE isolated from hiPSC-derived retinal organoids (RPE spheroids) and human donor RPE (Figure 1E). Similar to the results obtained with ARPE-19 cells, TRAP1 protein was also detected at the basal side in RPE spheroids (Figure 1F,G), where it co-localizes with TOM20, a mitochondrial marker (Figure 1G). This result is in accordance with previous electron microscopy studies on RPE spheroids in which mitochondria were localized at the basal side [26].

### 3.2. Impact of Oxidative Stress on TRAP1 Levels

Exposure to H_2_O_2_ is commonly used as a model to assess the oxidative stress susceptibility and antioxidant activity of RPE cells [6,7,33]. To define the optimal dose and incubation time, we exposed ARPE-19 cells to different concentrations of H_2_O_2_ for 2, 4 and 24 h and assessed the cell viability using the resazurin assay (Figure 2A and Appendix A). As expected, H_2_O_2_ decreased the viability of ARPE-19 cells in a dose-dependent manner (Figure 2A). For subsequent experiments, 500 μM H_2_O_2_ (24 h incubation) was used as these conditions decreased the cell viability of ARPE-19 cells by 53.5% ± 6.6% compared with the control (non-treated cells; *p =* 0.0286). To assess if the levels and localization of TRAP1 in ARPE-19 cells changed upon oxidative stress, cells were incubated with 500 μM of H_2_O_2_ for 24 h (Figure 2). Our results showed that the localization of TRAP1 was similar in both non-treated (Figure 2B) and treated cells (Figure 2C). However, the western blotting analysis demonstrated that TRAP1 levels decreased (61.4% ± 9.1%) upon exposure to 500 µM H_2_O_2_ for 24 h (Figure 2D,E).

### 3.3. TRAP1 Silencing Decreases Cell Metabolic Activity and Increases Reactive Oxygen Species Production

We wondered if TRAP1 silencing affects cell metabolism (measured as dehydrogenase activity), production of reactive oxygen species (ROS), proliferation and survival of RPE cells. The levels of TRAP1 were reduced by approximately 50% at 72 h upon silencing TRAP1 expression using specific siRNAs (siTRAP1) (Figure 3A,C–E) [30]. In contrast, the transfection of a control scramble RNA (siCTL), used as a control, did not affect TRAP1 levels (Figure 3A,B,D,E). Seventy-two hours after TRAP1 silencing, we detected a decrease of 33.8% in cell metabolic activity as measured by the resazurin assay (Figure 3F). Moreover, TRAP1 silencing did not exacerbate the decrease in cell metabolic activity in ARPE-19 cells exposed to 500 μM of H_2_O_2_ for 24 h (Figure 3G). The total number of cell nuclei (Figure 3H), cell mass (measured by Sulforhodamine B (SRB) assay (Figure 3I)), and the number of proliferating cells (measured by the number of Ki67-positive cells (Figure 3J)) remained similar to the control (siCTL). The data suggest that TRAP1 silencing promotes a decrease in cell metabolic activity in live cells, rather than an increase in cell death or a reduction in proliferation. 

To assess the effect of TRAP1 silencing on free radical levels, we evaluated the levels of superoxide anion production by dihydroethidium (DHE) assay at 24 and 72 h after transfection (Figure 4). Upon TRAP1 silencing, we observed an increase in DHE signal 24 and 72 h after transfection (Figure 4A–C and Figure 5D–F), suggesting that TRAP1 may regulate superoxide anion levels in these cells. 

### 3.4. TRAP1 Silencing Impairs Mitochondrial Structure

We have characterized the outcome of TRAP1 silencing on mitochondrial physiology. We started by assessing the impact on the mitochondrial ultrastructure by transmission electron microscopy (TEM). TRAP1 silencing in ARPE-19 cells had no major effect on the mitochondrial ultrastructure, in which clear cristae and membranes were observed (Figure 5A,B). To complement the ultrastructural analysis, mitochondria were stained with Image-iT™ TMRM Reagent (TMRM), and morphological parameters such as mitochondrial interconnectivity and elongation were quantitated [27]. No fragmentation of mitochondria was observed in siTRAP1 cells compared with the control (Figure 5C,D). Moreover, silencing of TRAP1 did not affect the mitochondrial interconnectivity (Figure 5E) but increased mitochondrial elongation (Figure 5F) compared with siCTL cells. The effects of TRAP1 silencing on ARPE-19 cells mitochondrial oxygen consumption rates (OCR) and extracellular acidification rate (ECAR) were also evaluated (Figure 6A). TRAP1 silencing significantly reduced the basal (40.1%) and maximal (42.8%) OCR (Figure 6 B,C) as well as the ATP production-linked (36.2%) and proton leak-related (14.6%) OCR (Figure 6E,F). Furthermore, the decrease in mitochondrial OCR was paralleled by a slight decrease in basal extracellular acidification rate (ECAR) (Figure 6D). Measurement of the two major energy-producing pathways of the cell, mitochondrial respiration (measured as the OCR) and glycolysis (indirectly measured as the ECAR) can reveal the four bioenergetic phenotypes that a cell can display: (a) quiescent; (b) energetic; (c) aerobic; and (d) glycolytic. Interestingly, by plotting the average data of basal OCR (Figure 6B) against basal ECAR (Figure 6D), we observed that TRAP1 silencing shifted the cell’s metabolic profile towards a more quiescent phenotype compared with untreated cells (Figure 6B,D).

## 4. Discussion

Tumor necrosis factor receptor-associated protein 1 (TRAP1) is expressed in several tissues and cell types [6,20,25,34,35,36,37,38,39]. TRAP1 mRNA is reported to be variably expressed in brain, skeletal muscle, heart, kidney, liver, lung, placenta and pancreas [34]. Previous studies also described that TRAP1 protein is expressed in several cell lines, such as H1299 human non-small cell lung carcinoma [20,25], G361 melanoma [34], PC-3M prostate carcinoma [35], A549 human adenocarcinoma [20,25,36,37], SW480 colorectal adenocarcinoma [34], HL-60 promyelocytic leukemia [34], SH-SY5Y neuroblastoma [38], HCT116 colon cancer [6] and human cervix carcinoma HeLa cell lines [39]. However, to the best of our knowledge, the expression of TRAP1 has never been described before in RPE, in contrast to its family member HSP90, which was already described as present in this tissue and the retina [40,41,42]. In the present study, we showed that TRAP1 is present in the mitochondria of human RPE cells and that TRAP1 silencing in ARPE-19 cells did not affect the number of cells. However, others have demonstrated that TRAP1 knockdown inhibits cell survival in different lung cancer cells, H1299 (human non-small cell lung carcinoma cell) and A549 (adenocarcinomic human alveolar basal epithelial cells) [20,25]. In our experiments, TRAP1 silencing also did not decrease cell proliferation. This is in contrast with results reported by Agorreta et al., and Palladino et al., who showed that downregulation of TRAP1 arrests cell proliferation in the A549 lung cancer cell line and thyroid carcinoma cells [36,37]. The discrepancy might be related to the levels of TRAP1 ablation or to the cell type itself.

After demonstrating the expression of TRAP1 in the RPE, we also analyzed if hydrogen peroxide-induced damage changed the TRAP1 levels and localization. We observed that TRAP1 levels were decreased 24 h after exposure to 500 µM of H_2_O_2_, without affecting the localization of the protein (Figure 2). However, others have demonstrated that TRAP1 expression is up-regulated following treatment with H_2_O_2_ (300 µM) for 24 h in a neuroblastoma cell line (SH-SY5Y) [38]. The discrepancy in the effect of H_2_O_2_ treatment in TRAP1 levels might be related to the cell type itself, since several studies show that the role of TRAP1 might differ depending on the cell type [20,36,39].

Our data showed that TRAP1 silencing results in increased ROS production (Figure 4), suggesting that TRAP1 plays an important role in regulating ROS production in RPE cells. Increased ROS production upon TRAP1 silencing was also previously reported in colon cancer cells (HCT116) [6], lung cancer cells [20] and HeLa cells upon exposure to H_2_O_2_ [39]. Finally, we also assessed the impact of TRAP1 silencing on mitochondria morphology. TRAP1 silencing led to increased mitochondrial elongation without affecting mitochondrial interconnectivity (Figure 5). Previous studies performed in MRC-5 cells demonstrated that cells with reduced levels of TRAP1 did not display alterations in mitochondrial morphology compared with the control cells and presented a complex interconnected mitochondrial network [20]. However, the same authors demonstrated that in A549 cells, TRAP1 silencing reduced mitochondrial elongation and interconnectivity [20]. It is known that TRAP1 plays a role in the regulation of mitochondrial energy metabolism, including oxidative phosphorylation (OXPHOS) processes, by interacting with proteins of the mitochondrial electron transport chain (ETC) complex II and complex IV [10]. Furthermore, we assessed the role of TRAP1 in the metabolic pattern and abilities of ARPE-19 cells. Our data showed that decreased levels of TRAP1 led to an overall decrease in respiratory capacity, driving cells into a quiescent state (Figure 6). Similarly, others demonstrated that A549 siTRAP1 cells presented decreased oxygen consumption without altered glycolytic rates [20]. In contrast, TRAP1 knockout murine adult fibroblast cells displayed a higher basal oxygen consumption rate and a significantly higher maximum respiratory capacity than control cells [36]. Recently, Xiao et al. also demonstrated that TRAP1 overexpression, but not TRAP1 ablation, increased maximal respiration and ATP production in cancer-associated fibroblasts [39]. 

In summary, our study demonstrates that TRAP1 silencing increases ROS production by human RPE cells and drives RPE cells into a quiescent metabolic state (Figure 7). Since oxidative-stress-induced RPE damage and mitochondrial dysfunction in RPE are hallmarks of AMD [7], it is plausible to speculate that TRAP1 might play a role in AMD pathology, opening new avenues for the development of therapeutical approaches based on the modulation of TRAP1. Therefore, future experiments to test the efficacy of *TRAP1* gene augmentation therapy in animal models of AMD are needed [43]. 

## Figures and Tables

**Figure 1 antioxidants-12-00381-f001:**
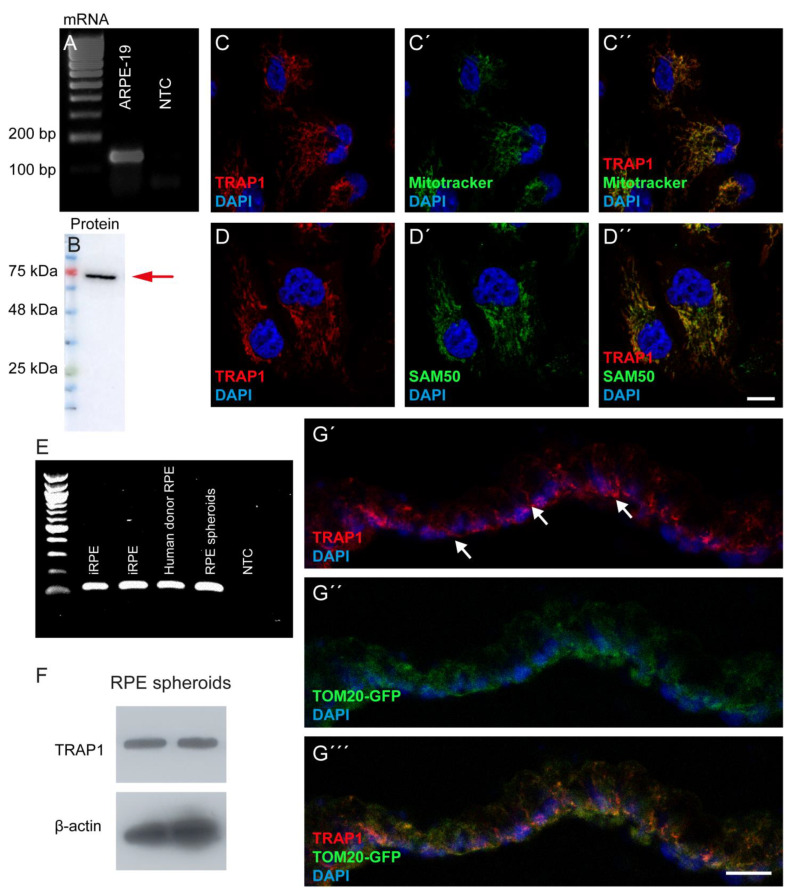
TRAP1 is expressed in RPE cells. TRAP1 mRNA (**A**) and protein (**B**) are expressed in human ARPE-19 cells. TRAP1 is located mainly in the mitochondria of ARPE-19 cells where it co-localizes with MitoTracker green (**C**) and the mitochondrial protein SAM50 (**D**). TRAP1 mRNA (**E**) is also expressed in iPSC-derived RPE cells (iRPE, two different lines), in RPE from human donors and in RPE spheroids (**E**). TRAP1 protein is also present in protein extracts from RPE spheroids (two different lines) (**F**) and co-localizes with the mitochondrial marker TOM20-GFP (**G**). NTC: Non-template control. Scale bars: 10 µm.

**Figure 2 antioxidants-12-00381-f002:**
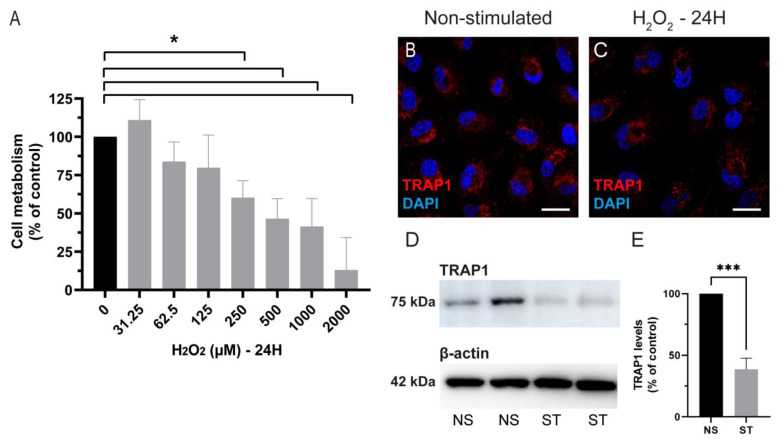
TRAP1 levels decrease upon challenge with hydrogen peroxide. ARPE-19 cells were treated with different doses of H_2_O_2_ (0; 31.25; 62.5; 125; 125; 250; 500; 1000 and 2000 μM) for 24 h (**A**). The dose of 500 μM was selected for the following experiments. ARPE-19 cells were treated, or not, with 500 μM H_2_O_2_ for 24 h (**B**–**E**). Scale bars in (**B**,**C**): 20 µm. Exposure to 500 μM H_2_O_2_ for 24 h resulted in lower levels of TRAP1. NS: Non-stimulated (black bars); ST: stimulated with H_2_O_2_ (grey bars). The values are presented as a percentage of the control (non-treated cells). Cell metabolism was assessed using the resazurin assay. Four independent experiments were performed (*n* = 4). For the graph (**A**), statistical significance was calculated by using the Mann–Whitney U test. For the graph (**E**), statistical significance was calculated by using an unpaired t-test. Statistically significant values: * *p* < 0.05, *** *p* < 0.001. In the graphs, all values are expressed as mean ± standard error of the mean (SEM).

**Figure 3 antioxidants-12-00381-f003:**
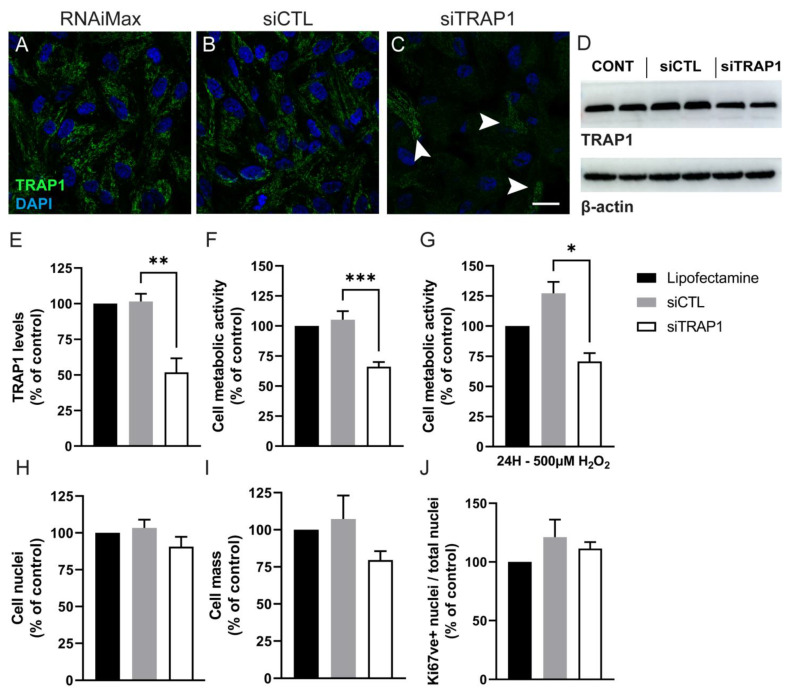
TRAP1 silencing decreases ARPE-19 cellular metabolism. Seventy-two hours after TRAP1 silencing using specific siRNAs (siTRAP1), the levels of TRAP1 were reduced by approximately 50% (**A**,**C**–**E**). Some cells still expressed high levels of TRAP1 (C, arrowheads).The scramble (control) RNA (siCTL) did not affect TRAP1 levels (**A**,**B**–**E**) compared with the incubation with only the transfection reagent (lipofectamine RNAiMAX) (**A**,**D**,**E**). Scale bars in (**A**–**C**): 20 µm. TRAP1 siRNA-mediated silencing decreased the cell metabolic activity as measured by the resazurin assay (**F**). TRAP1 silencing did not further decrease cell metabolic activity upon treatment of cells with 500 µM H_2_O_2_ for 24 h (**G**). The total number of cells (**H**), cell mass measured by SRB assay (**I**) and the number of proliferating Ki67-positive cells (**H**) did not change upon TRAP1 silencing. Black bars: cells incubated with lipofectamine RNAiMAX reagent only. Grey bars: cells transfected with lipofectamine RNAiMAX reagent and siCTL. White bars: cells transfected with lipofectamine RNAiMAX reagent and siTRAP1. Number of independent experiments: (**E**) *n* = 5; (**F**) *n* = 8; (**G**) *n* = 4; (**H**) *n* = 5; (**I**) *n* = 6; (**J**) *n* = 4. Statistical significance was calculated by using the Mann–Whitney U test. Statistically significant values: * *p* < 0.05, ** *p* < 0.01, *** *p* < 0.001. In the graphs, all values are expressed as mean ± standard error of the mean (SEM).

**Figure 4 antioxidants-12-00381-f004:**
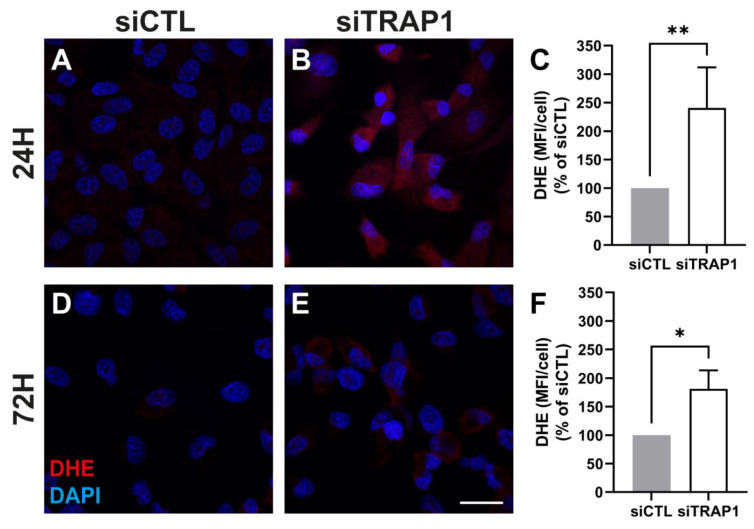
TRAP1 silencing in ARPE-19 cells increase the levels of superoxide anion. Representative confocal microscopy images of fluorescent detection of ROS by dihydroethidium (DHE) in cells transfected with siCTL (**A**,**D**) and siTRAP1 (**B**,**E**) for 24 h (**A**,**B**) and 72 h (**D**,**E**). Scale bars: 20 µm. Mean fluorescence intensity (MFI) of at least 4 images per condition/experiment was measured using Image J v1.53C and divided by the number of the DAPI-positive nuclei (MFI/cell) (**C**,**F**). Four independent experiments were performed (*n* = 4). Statistical significance was calculated by using the Mann––Whitney U test. Statistically significant values: * *p* < 0.05, ** *p* < 0.01. In the graphs, all values are expressed as mean ± standard error of the mean (SEM).

**Figure 5 antioxidants-12-00381-f005:**
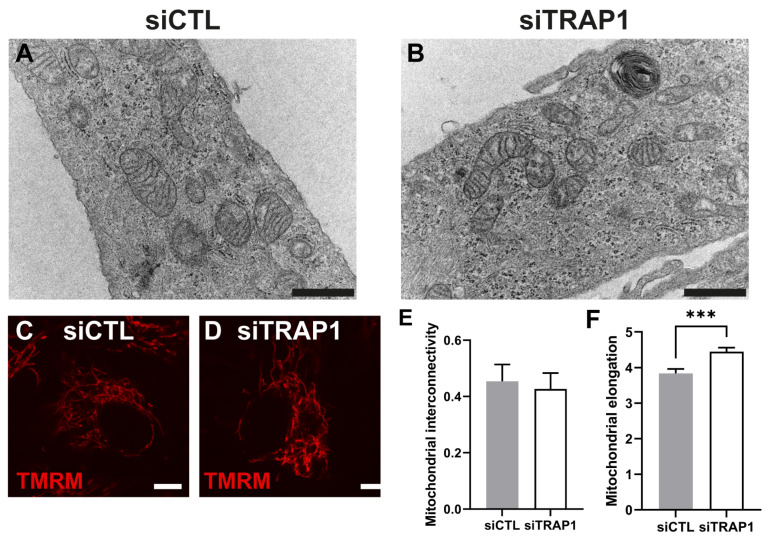
TRAP1 silencing increases mitochondrial elongation without affecting mitochondrial ultrastructure. Transmission electron microscopy (TEM) images of mitochondria from ARPE-19 cells transfected with siCTL (**A**) or siTRAP1 (**B**). Representative mitochondria are presented. Scale bars in A and B: 100 nm. Analysis of confocal microscopy images obtained using Image-iT™ TMRM Reagent (TMRM) for mitochondrial labelling in ARPE-19 cells transfected with siCTL (**C**) or siTRAP1 (**D**). Scale bars in C and D: 50 µm. The mitochondrial interconnectivity was similar between the two experimental groups (**E**). siTRAP1 cells showed increased mitochondrial elongation (**F**). Statistical significance was calculated by using an unpaired t-test. Statistically significant values: *** *p* < 0.001. In the graphs, all values are expressed as mean ± standard error of the mean (SEM).

**Figure 6 antioxidants-12-00381-f006:**
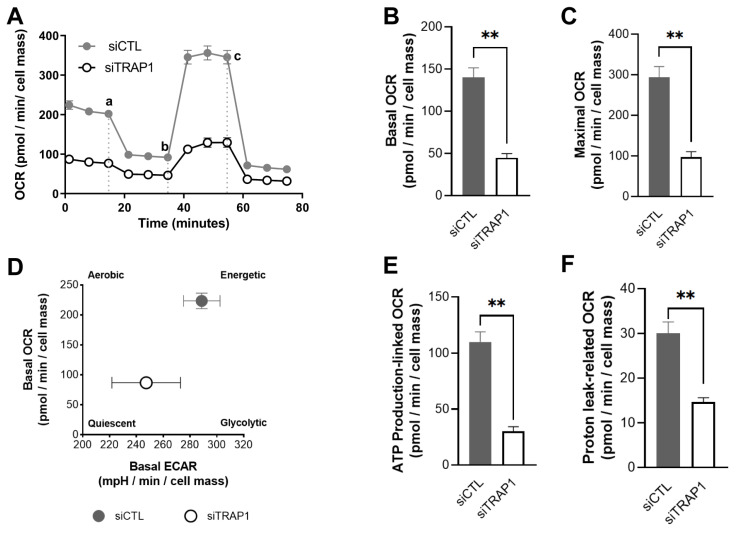
TRAP1 silencing drives a quiescent metabolic state. Representative images of oxygen consumption rate (OCR) and extracellular acidification rate (ECAR) measurements using a Seahorse XFe96 Extracellular Flux Analyzer (**A**–**F**). siTRAP1 cells presented a decreased basal and maximal OCR compared with the siCTL cells (**B**,**C**). A shift towards a more quiescent phenotype was observed when the average mitochondrial basal OCR was plotted against the average basal ECAR (**D**). A decrease in mitochondrial OCR was paralleled by a slight decrease in ECAR (**D**). siTRAP1 silencing significantly reduced the ATP production-linked (**E**) and proton leak-related (**F**) OCR. Data are the mean ± SEM of 3 independent experiments and show the effects of mitochondrial inhibitors (a) oligomycin (2 µM), (b) FCCP (1 µM) and (c) antimycin A (1 µM) plus rotenone (1 µM) injected as indicated. The results are expressed as pmol O_2_/min/cell mass for OCR and mpH/min/cell mass for ECAR. Statistical significance was calculated by using an unpaired *t*-test. Statistically significant values: ** *p* < 0.01. In the graphs, values are expressed as mean ± standard error of the mean (SEM).

**Figure 7 antioxidants-12-00381-f007:**
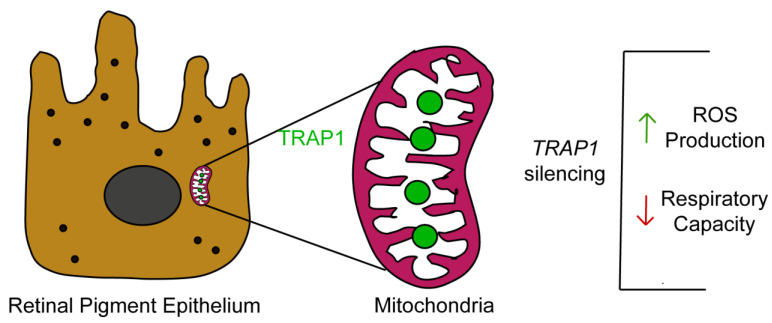
Graphical representation of the highlights of our study. TRAP1 is present in retinal pigment epithelial cells and is located mainly in the mitochondria. TRAP1 silencing increases reactive oxygen species (ROS) production and decreases mitochondrial respiratory capacity.

## Data Availability

The data are contained within the article and Appendix A.

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
