# Peer review of "TRAP1 Is Expressed in Human Retinal Pigment Epithelial Cells and Is Required to Maintain their Energetic Status"

_antioxidants, 2023, doi:10.3390/antiox12020381_

Round 1
Reviewer 1 Report
The authors show the expression of TRAP1(tumor necrosis factor receptor-associated protein1) in PPE cells using various methods and its role. Since TRAP1 is a potential target of AMD, I also believe that the results should impact the development of a new drug for AMD. The overall quality of the manuscript is high enough to publish in Antioxidant. I recommend it to accept publication.
Author Response
We thank reviewer 1 for his positive comments.
Reviewer 2 Report
This manuscript, submitted by Rego et. al, “TRAP1 is expressed in human retinal pigment epithelial cells and is required to maintain their energetic status”, assessed the presence and the function of TRAP1 in human RPE. Study design is reasonable. It is the first study to investigate expression of TRAP1 in ocular. Moreover, this study demonstrated that TRAP1 has antioxidant functions in RPE cells. The results are of interest. This study can open the avenue to further study the role of TRAP1 in RPE-related ocular diseases. However, there are some issues needed to be addressed before publication.
1. Line 359-361, the authors mentioned “to define the optimal dose and incubation time”. However, we couldn’t see the results for various timepoints, although we saw the results for different concentration of H2O2. The author shall address this issue. 2. What is the error bar for the graph A and C in Figure 1, SEM or SD? In addition, what was the statistical analysis used for comparison, unpaired t-test or the Mann-Whitney U test? Such issue also occurred in the other Figures. Suggest the authors add the information in Figure Legends.
3. In discussion section, the authors shall add information about the distribution of TRA in other cells and tissues reported by previous studies.
4. TRAP1 levels were decreased after exposure to H2O2 in this study, while their levels up-regulated in other cells in previous studies. What are the reasons for the discrepancy between their study and others? The authors shall add some comments in the discussion section.
Author Response
- Line 359-361, the authors mentioned “to define the optimal dose and incubation time”. However, we couldn’t see the results for various timepoints, although we saw the results for different concentration of H2O2. The author shall address this issue.
Answer: Taking into consideration the comment of reviewer 2, we added a supplementary figure with the data that refers to the testing of different timepoints, not only 24 hours but also 2 and 4 hours. We have also adjusted the text accordingly.
“To define the optimal dose and incubation time, we exposed ARPE-19 cells to different concentrations of H2O2 for 2, 4 and 24 hours and assessed the cell viability using the resazurin assay (Fig. 2A and Fig.S1).”
“Figure S1. Assessment of cell metabolism of ARPE-19 cells upon challenge with hydrogen peroxide. ARPE-19 cells were treated with different doses of H2O2 (0; 31.25; 62.5; 125; 125; 250; 500; 1000 and 2000 μM) for 2 hours (A) and 4 hours. NS: Non-stimulated (black bars); ST: stimulated with H2O2 (grey bars). The values are presented as a percentage of the control (non-treated cells). Cell metabolism was assessed using the resazurin assay. Four independent experiments were performed (n=4). Statistical significance was calculated by using the Mann-Whitney U test. Statistically significant values: * p < 0.05. In the graphs, all values are expressed as mean ± standard error of the mean (SEM).”
- What is the error bar for the graph A and C in Figure 1, SEM or SD? In addition, what was the statistical analysis used for comparison, unpaired t-test or the Mann-Whitney U test? Such issue also occurred in the other Figures. Suggest the authors add the information in Figure Legends.
Answer: Although there are no graphs in Figure 1, taking into consideration the comment of reviewer 2, we adapted all the figure legends by adding the information of the test used to calculate statistical significance.
“Figure 2. […] For the graph A, statistical significance was calculated by using the Mann-Whitney U test. For the graph E, statistical significance was calculated by using an unpaired t-test. Statistically significant values: * p < 0.05, ** p < 0.01, *** p < 0.001. In the graphs, all values are expressed as mean ± standard error of the mean (SEM).”
“Figure 3. […] Statistical significance was calculated by using the Mann-Whitney U test. Statistically significant values: * p < 0.05, ** p < 0.01, *** p < 0.001. In the graphs, all values are expressed as mean ± standard error of the mean (SEM).”
“Figure 4. […] Statistical significance was calculated by using the Mann-Whitney U test. Statistically significant values: * p < 0.05, ** p < 0.01. In the graphs, all values are expressed as mean ± standard error of the mean (SEM).”
“Figure 5. […] Statistical significance was calculated by using an unpaired t-test. Statistically significant values: * p < 0.05, ** p < 0.01, *** p < 0.001. In the graphs, all values are expressed as mean ± standard error of the mean (SEM).”
“Figure 6. […] Statistical significance was calculated by using an unpaired t-test. Statistically significant values: * p < 0.05, ** p < 0.01, *** p < 0.001. In the graphs, values are expressed as mean ± standard error of the mean (SEM).”
“Figure S1. […] Statistical significance was calculated by using the Mann-Whitney U test. Statistically significant values: * p < 0.05. In the graphs, all values are expressed as mean ± standard error of the mean (SEM).”
- In discussion section, the authors shall add information about the distribution of TRA in other cells and tissues reported by previous studies.
Answer: We added the requested information to the text: “Tumour necrosis factor receptor-associated protein 1 (TRAP1) is expressed in several tissues and cell types [6,20,25,34,35,39-42]. TRAP1 mRNA is reported to be variably expressed in brain, skeletal muscle, heart, kidney, liver, lung, placenta and pancreas [34]. Previous studies also described that TRAP1 protein is expressed in several cell lines, such as H1299 human non-small cell lung carcinoma [20,25], G361 melanoma [34], PC-3M prostate carcinoma [35], A549 human adenocarcinoma [20,25,39,40], SW480 colorectal adenocarcinoma [34], HL-60 promyelocytic leukemia [34] , SH-SY5Y neuroblastoma [41], HCT116 colon cancer [6] and human cervix carcinoma HeLa cell lines [42]. However, to the best of our knowledge, the expression of TRAP1 was never described before in RPE in contrast to its family member HSP90, which was already described as present in this tissue and the retina [36-38]. In the present study, we showed that TRAP1 is present in the mitochondria of human RPE cells and that TRAP1 silencing in ARPE-19 cells did not affect the number of cells.”
- TRAP1 levels were decreased after exposure to H2O2 in this study, while their levels up-regulated in other cells in previous studies. What are the reasons for the discrepancy between their study and others? The authors shall add some comments in the discussion section.
Answer: We added the requested information to the text: ““However, others, demonstrated that TRAP1 expression is up-regulated following treatment with H2O2 (300 µM) for 24 hours in a neuroblastoma cell line (SH-SY5Y) [41]. The discrepancy in the effect of H2O2 treatment in TRAP1 levels might be related to the cell type itself, since several studies show that TRAP1 role might differ depending on the cell type [20,39,42].
Reviewer 3 Report
Authors carried out an interesting study about the role of TRAP1 in mitochondrial activity of retinal pigmented epithelial cells.
However the manuscript must be throughfully revised in order to improve the overall quality of the study, due to the following issues.
- authors must improve introduction, since I did not catch the link between trap1 and AMD, authors must spend more words in the introduction about the experimental hypothesis.
- statistical analysis is not proper, authors reported mean+/-SEM for samples (technical or independent samples?) n=3. SEM must be used for n>7.
- for clarity I would shift figure 5 in two figures
- I would add a figure resuming the role of TRAP1 in ARPE-19 cells
- Discussion cover most of results, however, similarly to the introduction the link to AMD is not totally clear. Moreover authors must properly rephrase the final sentence lines 507-509, because AMD is a multifactorial diseases that would affect patients in different forms (dry AMD, wet AMD). Do the authors hypothesize a gene therapy approach? Or the use of any molecule able to increase TRAP1 expression? Please discuss about this issue.
Provide the tracked version of the revised manuscript
Author Response
- authors must improve introduction, since I did not catch the link between trap1 and AMD, authors must spend more words in the introduction about the experimental hypothesis
Answer: We are grateful for the comment that helps improving this section. Taking into consideration the comment of reviewer 3, we adapted our text to explain the link between TRAP1 and AMD and to clarify our experimental hypothesis.
“[…] Degeneration and impaired clearance mechanisms of RPE contribute to increased accu-mulation of lipofuscin, a yellowish aggregate of oxidized proteins and lipids, the main constituent of drusen, one of the hallmarks of AMD. Therefore, RPE has been suggested as a critical site of pathology [6]. Retinal pigment epithelial cells are particularly susceptible to oxidative stress due to exposure to intense focal light, high metabolic activity, unique phagocytic function, large oxygen gradient from the choroid, and accumulation of oxi-dized lipoproteins with ageing [6]. In the context of AMD, Since oxidative stress-induced RPE damage results from the excessive accumulation of reactive oxygen species (ROS), produced mainly in mitochondria under pathological conditions. M, mitochondrial func-tion homeostasis is critical for maintaining ROS at physiological levels, avoiding meta-bolic dysfunction observed in AMD pathology [7] [7]. Therefore, it is important to study novel proteins that might be involved in oxidative stress mechanisms in the RPE in the context of AMD. Tumour necrosis factor receptor-associated protein 1 (TRAP1), also known as heat shock protein 75 (HSP75), is a mitochondrial molecular chaperone that supports protein folding and contributes to the maintenance of mitochondrial integrity even under cellular stress [8-10]. TRAP1 has attracted increasing interest for its homology to heat shock protein 90 (HSP90), long pursued as a potential therapeutic target for de-signing novel anticancer agents [9,11-13]. Previous studies in several cell lines demon-strated that TRAP1 protects against mitophagy, mitochondrial apoptosis, and dysfunc-tion by decreasing the production and accumulation of ROS [8,14-17]. It has also been suggested that downregulation of TRAP1 expression impacts cellular function, resulting in lower mitochondrial membrane potential, increased intracellular ROS production and increased cell death [18-20]. TRAP1 inhibits mitochondrial oxidative phosphorylation (OXPHOS) via downregulation of cytochrome oxidase in the respiratory chain [9]. Con-sidering the above mentioned TRAP1 functions described in other cell types [8-20], we hypothesize that TRAP1 modulates mitochondrial metabolism in the RPE, and has a role in avoiding AMD onset and progression. […]”
- statistical analysis is not proper, authors reported mean+/-SEM for samples (technical or independent samples?) n=3. SEM must be used for n>7.
Answer: In our results, n=3 means that we evaluated 3 independent samples, with 3 technical replicates each. In our results, we report mean +/- SEM. Additionally, SEM or SD is only a matter of presenting the results, since they are independent from the statistical test applied and do not interfere with the result of the statistical test.
- for clarity I would shift figure 5 in two figures
Answer: We appreciate the thoughtful suggestion. Taking into consideration the comment of reviewer 3, we have split figure 5 into two figures:
“Figure 5. TRAP1 silencing increases mitochondrial elongation without affecting mitochondrial ultrastructure. Transmission electron microscopy (TEM) images of mitochondria from ARPE-19 cells transfected with siCTL (A) or siTRAP1 (B). Representative mitochondria are presented. Scale bars in A and B: 100 nm. Analysis of confocal microscopy images obtained using Image-iT™ TMRM Reagent (TMRM) for mitochondrial labelling in ARPE-19 cells transfected with siCTL (C) or siTRAP1 (D) Scale bars in C and D: 50 µm. The mitochondrial interconnectivity was similar between the two experimental groups (E). siTRAP1 cells showed increased mitochondrial elongation (F). Statistical significance was calculated by using an unpaired t-test. Statistically significant values: * p < 0.05, ** p < 0.01, *** p < 0.001. In the graphs, all values are expressed as mean ± standard error of the mean (SEM).”
“Figure 6. TRAP1 silencing drives a quiescent metabolic state. Representative images of oxygen consumption rate (OCR) and extracellular acidification rate (ECAR) measurements using a Seahorse XFe96 Extracellular Flux Analyzer (A-F). siTRAP1 cells presented a decreased basal and maximal OCR when compared to the siCTL cells (B,C). A shift towards a more quiescent phenotype was observed when the average mitochondrial basal OCR was plotted against the average basal ECAR (D). A decrease in mitochondrial OCR was paralleled by a slight decrease in ECAR (D). siTRAP1 silencing significantly reduced the ATP production-linked (E) and proton leak-related (F) OCR. Data are the mean ± SEM of 3 independent experiments and show the effects of mitochondrial inhibitors (a) oligomycin (2 µM), (b) FCCP (1 µM) and (c) antimycin A (1 µM) plus rotenone (1 µM) injected as indicated. The results are expressed as pmol O2/min/cell mass for OCR and mpH/min/cell mass for ECAR. Statistical significance was calculated by using an unpaired t-test. Statistically significant values: * p < 0.05, ** p < 0.01, *** p < 0.001. In the graphs, values are expressed as mean ± standard error of the mean (SEM).”
- I would add a figure resuming the role of TRAP1 in ARPE-19 cells
Answer: Taking into consideration the suggestion of reviewer 3, we added a figure resuming the highlight findings of our study on the role of TRAP1 in ARPE-19 cells.
“Figure 7. Graphical representation of the highlights of our study. TRAP1 is present in retinal pigment epithelial cells and is located mainly in the mitochondria. TRAP1 silencing increases reactive oxygen species (ROS) production and decreases mitochondrial respiratory capacity.”
- Discussion cover most of results, however, similarly to the introduction the link to AMD is not totally clear. Moreover authors must properly rephrase the final sentence lines 507-509, because AMD is a multifactorial diseases that would affect patients in different forms (dry AMD, wet AMD). Do the authors hypothesize a gene therapy approach? Or the use of any molecule able to increase TRAP1 expression? Please discuss about this issue
Answer: Again, we thank the reviewer for the useful comment, that allowed a clear improvement in this section. Taking into consideration the comment of reviewer 3, we made several adjustments to our text. We have clarified the link between TRAP1 functions in the RPE and AMD. We have rephrased the final sentence as suggested. We have also discussed possible future approaches concerning gene therapy and/or the study of TRAP1 overexpression in disease models of dry and wet AMD.
“In summary, our study demonstrates that TRAP1 silencing increases ROS production by human RPE cells and drives RPE cells into a quiescent metabolic state (Fig.7). Since oxidative stress-induced RPE damage and mitochondrial disfunction in RPE are hallmarks of AMD [7], it is plausible to speculate that TRAP1 might play a role in AMD pathology, opening new avenues for the development of therapeutical approaches based on the modulation of TRAP1. Therefore, future experiments to test the efficacy of TRAP1 gene augmentation therapy in animal models of AMD are needed [43].”